



# Results from the 4[th] WMO Filter Radiometer Comparison for aerosol optical depth measurements

Stelios Kazadzis[1,11], Natalia Kouremeti[1], Henri Diémoz[2], Julian Gröbner[1], Bruce W. Forgan[3], Monica Campanelli[4], Victor Estellés[5], Kathleen Lantz[6], Joseph Michalsky[6], Thomas Carlund[7], Emilio Cuevas[8], Carlos Toledano[9], Ralf Becker[10], Stephan Nyeki[1], Panagiotis G. Kosmopoulos[11], Viktar Tatsiankou[12], Laurent Vuilleumier[13], Frederick. M. Denn[14], Nozomu Ohkawara[15], Osamu Ijima[15], Philippe Goloub[16], Panagiotis I. Raptis[11,1], Michael Milner[3], Klaus Behrens[10], Africa Barreto[8,9,17], Giovanni Martucci[13], Emiel Hall[6], James Wendell[6], Bryan E. Fabbri[14], Christoph Wehrli[1]

[1]Physikalisch-Meteorologisches Observatorium Davos, World Radiation Center, Switzerland

[2]Aria e Atmosfera - Radiazione solare e atmosfera ARPA Valle, Saint-Christophe 11020, Italy

[3]Standards & Metrology, Bureau of Meteorology, Docklands Vic 3008, Australia

[4]SACI-CNR,Via Fosso del Cavaliere 100, 00133, Rome, Italy

[5]Department of Earth Physics and Thermodynamics, Solar Radiation and Research Unit, Univ. de València, Valencia, Spain

[6]Cooperative Institute for Research in Environmental Studies, NOAA/ESRL/GMD, Boulder, CO 80305, U.S.A.

[7]Swedish Meteorological and Hydrological Institute, 601 76 Norrköping, Sweden

[8]Izaña Atmospheric Research Centre, State Meteorological Agency (AEMET), Santa Cruz de Tenerife, Spain

[9]Atmospheric Optics Group (GOA), University of Valladolid 47011, Valladolid, Spain

[10]Deutscher Wetterdienst Meteorologisches Observatorium Lindenberg, D-15848 Tauche, Germany

[11]Institute of Environmental Research and Sustainable Development, National Observatory of Athens, Greece

[12] COFOVO Energy Inc. 800 King Edward Avenue, Suite 3014 Ottawa, ON, K1N 6N5 Canada

[13] Federal Office of Meteorology and Climatology MeteoSwiss, Payerne, Switzerland

[14] Science Systems & Applications Inc NASA Langley Science Directorate, Hampton VA 23666, U.S.A

[15] Japan Meteorological Agency 1-3-4 Otemachi, Chiyoda-ku, 100-8122 Tokyo, Japan

[16] Laboratoire d' Optique Atmosphérique, Univ. des Sciences et Technologies de Lille 159655 Villeneuve d'Ascq, France

[17]Cimel Electronique, 75011, Paris, France

*Correspondence to:* S. Kazadzis (stelios.kazadzis@pmodwrc.ch)

**Abstract.**

This study presents the results of the 4[th] Filter Radiometer Comparison that was held in Davos, Switzerland, between September 28 and October 16, 2015. Thirty filter radiometers and spectroradiometers from 12 countries participated



including reference instruments from global aerosol networks. The absolute differences of all instruments compared to the reference have been based on the WMO criterion defined as "95% of the measured data has to be within 0.005±0.001/m" (where m is the air mass). At least 24 out of 29 instruments achieved this goal at both 500 and 865 nm, while 12 out of 17 and 13 out of 21 achieved this at 368 and 412 nm, respectively. While searching for sources of differences among different

instruments, it was found that all individual differences linked to Rayleigh, $NO_2$, ozone, water vapor calculations and related optical depths and air mass calculations were smaller than 0.01 in AOD at 500 and 865 nm. Different cloud detecting algorithms used have been compared. Ångström exponent calculations showed relatively large differences among different instruments partly because of the sensitivity of this parameter at low AOD conditions. The overall low deviations of these AOD results and the high accuracy of reference aerosol network instruments demonstrated a promising framework to

achieve homogeneity, compatibility and harmonization among the different spectral AOD networks in the near-future.

**Keywords.** Aerosol optical depth; Filter Radiometer Comparison

**Introduction**

Growing recognition of the role of atmospheric aerosols in the determination and modification of the Earth's radiation budget and hydrological cycle through their direct and indirect effects has led to a steady increase of scientific interest in aerosol

physical, chemical and optical properties over the last decades (Augustine et al., 2008; Lohmann and Feichter, 2005; Nyeki et al., 2012; Wehrli, 2008). The main parameter related to columnar integrated optical activity of aerosols is their optical depth that can be derived from ground-based measurements of the attenuation of sunlight, but also with modeling of scattered radiation observed from space (Chylek et al., 2003; Shaw et al., 1973; Toledano et al., 2011). Aerosol optical depth (AOD) is the single most comprehensive variable to assess the aerosol load of the atmosphere and represents the least

common denominator by which ground based observations, satellite retrievals, and global modeling of aerosol properties are compared, providing a holistic approach for an all-around understanding and quantification of the AOD uncertainties (Heintzenberg et al., 1996; Andrews et al., 2017). This significance is illustrated by the fact that AOD is one of the core aerosol parameters of the World Meteorological Organization (WMO, 2003) Global Atmosphere Watch (GAW) program.

AOD can be derived from the ground with measurements of the spectral transmission of direct solar radiation by various types of instruments such as sun-pointing or rotating shadow band filter radiometers, as well as spectroradiometers. It can be determined as the difference between the observed total optical depth and the modeled optical depths of molecular (Rayleigh) scattering and gaseous absorption, which depend on wavelength. Since AOD is often a small difference between two larger numbers (mainly the total optical depth and the Rayleigh scattering), it is very sensitive to small calibration errors

and to a lesser degree on the chosen algorithms for the modeled components. The main source of error in sun photometry is the use of incorrectly estimated calibration constants, $V_0(\lambda)$. The calibration constant, so-called exoatmospheric value, is the



signal or voltage, $V_0(\lambda)$ that the filter radiometers and spectroradiometers would measure in the absence of an attenuating atmosphere, as if it were measuring at the top of the atmosphere. The constant is commonly determined through Langley extrapolations, which can achieve a relative uncertainty of 1% or better (Schmid and Wehrli, 1995, Holben et al., 1998) in

the UVA to NIR spectral range (Neckel and Labs, 1984). The Langley method consists of performing sun photometer measurements at different solar elevation angles (or optical air masses) throughout a day under very stable atmospheric conditions and pristine skies, and plotting the logarithm of these voltages against the relative air mass. The determination of $V_0(\lambda)$ values by the Langley method has been the main current practice for calibration of spectral radiometers used in AOD observations. In addition, other in situ calibrations (Nakajima et al., 1996; Campanelli et al., 2004; 2007) have been

proposed. According to the Beer-Lambert-Bouguer law, the ordinate intercept yields the logarithm of the zero-air-mass photometer voltage $V_0(\lambda)$ if the turbidity of the atmosphere remains constant during the measurements (Dirmhirn et al., 1993). Langley extrapolation relies on the assumption of stable optical depth during the period of measurements. Standard least squares fitting techniques are applicable only under the additional assumption of a normal distribution of optical depth fluctuations. However, certain cases of systematic variations of the AOD can induce unnoticed systematic errors in the

calibration constant (Shaw, 1976), that may lead to a significant day-to-day scatter. Langley extrapolations are thus rarely successful at most observation sites and are usually performed at high altitude sites or at places where an additional independent assessment of AOD variations can be used. Although the stability of optical interference filters has improved a lot over the last 20 years, periodic re-calibrations of filter radiometers are still needed in order to maintain AOD uncertainties within certain limits.


Surface-based global networks of AOD measurements, such as the AErosol RObotic NETwork (AERONET) (Holben et al. 1998; 2001), the Global Atmospheric Watch Precision Filter Radiometer network (GAW-PFR) (McArthur et al., 2003; Wehrli, 2005), the SKYradiometer NETwork (SKYNET) (Aoki et al., 2006; Kim et al., 2008), the Bureau of meteorology AOD Australian network (BoM) (Mitchell and Forgan, 2003), and the National Oceanic and Atmospheric

Administration/Earth System Research Laboratory's (NOAA/ESRL) SURFRAD network (Augustine et al, 2000) and NOAA/ESRL Global Baseline Observatories (Dutton et al., 1994; NOAA/ESRL, 2003) are used to measure spectral AODs at various locations worldwide. Several AOD intercomparison campaigns with the participation of different instrument types that belong to some of the above networks have taken place as short-term intensive field campaigns and have proven themselves a successful method of relating the methodologies of standards from one network to another (Aoki et al., 2006;

Kim et al., 2005; McArthur et al., 2003; Mitchell and Forgan 2003; Schmid et al., 1999).
Simultaneously, most of the previous AOD comparison studies, including the 1st, 2nd and 3rd filter radiometer comparisons (FRC-I, -II and -III), were conducted under clear atmospheric conditions which are preferred for evaluating the differences in instrument calibrations. Results from FRC-I to III were not published as the intercomparisons were effectively organized on an *ad hoc* basis amongst participants of the International Pyrheliometer Comparisons (IPC) at the Physikalisch-

Meteorologisches Observatorium Davos, World Radiation Center (PMOD/WRC), Davos, Switzerland. FRC-I to -III were



held for 2 weeks in September – October of 2000, 2005, 2010, respectively. FRC-II and –III were based on AOD results derived from simultaneous measurements by each participant according to their standard protocol and evaluated by their preferred algorithms, including cloud-screening. Recommendations by WMO experts (WMO, 2005) were implemented as of FRC-II. A large number of radiometers were present during both FRC-II (14 from 9 countries) and FRC–III (17 from 10 countries). The main conclusions were that: i) Most of the ground-based AOD measuring instruments were able to achieve comparable results to within ≈±0.005, ii) algorithms used for calibration and evaluation contributed a significant fraction of the observed dispersion in AOD measurements, and iii) measurements of the Ångström exponent (AE) for the wavelength pair 500/862 nm were questionable when AOD < 0.1.  In this study, we present the results of the 4[th] FRC intercomparison campaign in which 30 instruments, from 12 countries, belonging to the above-mentioned global or national networks, participated. Section 2 presents the instrumentation, the location of measurements and the analytical methodology used. Section 3 describes the intercomparison results while conclusions in Section 4 investigates AOD calculation methods and involved assumptions and set the framework within which the homogeneity of networks will be feasible through standardization of instrumentation and procedures in combination with a multi-faceted data quality control /quality assurance system. The whole activity aims to homogenize/harmonize AOD measurements on a global scale. The comparison protocol was formulated according to the WMO recommendations (WMO, 2003; 2005).

## 2 Instrument, location and AOD retrieval

### 2.1 Intercomparison location

The World Optical depth Research and Calibration Center (WORCC) was established at Davos in 1996 and assigned the mission by WMO to develop stable instrumentation and improved methods of calibration and observation of AOD. These new developments were demonstrated in a global pilot network (Wehrli, 2008). Toward this goal and concurrent with the 12[th] International Pyrheliometer Comparisons (IPC-XII), FRC-IV was held. Instrumentation belonging to different aerosol optical depth global networks were invited to participate. The comparison took place on the premises of the PMOD/WRC from September 28 to October 16, 2015. Thirty filter radiometers and spectroradiometers from 12 countries participated in this campaign. PMOD/WRC (46° 49' N, 9° 51' E, 1590 m above sea level) is situated at the edge of the small town of Davos in the eastern part of Switzerland. The valley of Davos is oriented NorthEast – SouthWest and the horizon limits solar observations to zenith angles smaller than about 78° (from about 7:15 to 16:15 hours CET) in fall. Average sunshine duration in September and October is 173 and 156 hours, respectively, while average long term AOD is ~0.06 at 500 nm (Nyeki et al., 2012).



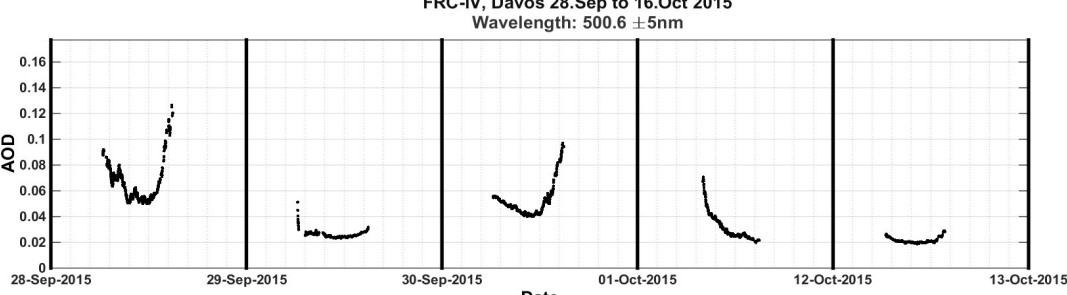


**Figure 1:** Average AOD at 500 nm measured by the WORCC triad during 5 days with cloud free sky conditions. The symbols represent 1-minute measurements.

During FRC-IV, there were five days (September 28 – 30, October 1 and 12) mainly with sunshine and only very limited

presence of clouds. Measurements from these days have been used to compare the participating instruments. During the five
intercomparison days, AOD varied from 0.02 up to 0.12 at 500 nm, which can be considered as normal values for the area.
Figure 1 shows the AOD variability during the intercomparison days, as measured by the WORCC triad that is defined as the
mean of three well-maintained PFR instruments. Before the start of the campaign, the PFR triad was inter-compared with
three additional PFR instruments that had performed measurements at Izaña, Tenerife, Spain, (2 instruments) and Mauna

Loa, Hawaii, USA, for a period of 9 months. The calibration of the particular instruments was based on the Langley
calibration technique. During five cloudless days in August - September 2015, the three Langley-calibrated instruments were
compared with the three PFR triad instruments. The differences in AOD for all instruments were from 0.2% up to 0.5% or up
to 0.0005 in AOD to at all wavelengths.

## 2.2 Participating instruments

Filter radiometers have been used in meteorology for at least 40 years to measure atmospheric haze or turbidity. Modern sun
photometers use dielectric interference filters and silicon photodetectors resembling the filter radiometers used in metrology.
The precision filter radiometers (PFRs; Wehrli, 2000), has been designed with emphasis on radiometric stability and a small
number of instruments were built for a trial network of AOD measurement sites (Wehrli, 2005).
    Thirty instruments from 12 countries participated in FRC-IV representing the most widely used instrument types for AOD

retrieval. The participating filter radiometers were either of the direct pointed type, e.g. classic sun photometers, including
sky-scanning radiometers used in direct sun mode, or hemispherical rotating shadow-band radiometers. These included the
following (see Table 1 for further details):

a.    Nine (9) instruments were of the PFR type (manufactured by PMOD/WRC) that is used in the GAW AOD network
    (Wehrli, 2005). The PFR is a classic sun photometer with 4 independent channels, a field-of-view (FoV) of 2.5° and



equipped with 3 to 5 nm bandwidth interference filters. The detector unit is held at a constant temperature of 20°C by an active Peltier system.

b. Two (2) radiometers were of the Carter-Scott SP02 type (Mitchell and Forgan, 2003), which is similar to the PFR, but has a wider FoV of 5° and no temperature controller.

c. Three (3) CIMEL CE318 sun and sky scanning radiometers as used by AERONET (Holben et al., 1998), two of them CE318-T model, the new standard AERONET instrument with improved performance and capable to perform lunar observations (Barreto et al., 2016). These instruments have a narrow FoV of 1.2° and sequentially measure the sun at 9 wavelengths within a few seconds. No temperature control is used.

d. Four (4) MFRSR rotating shadow band radiometers (Harrison et al., 1994; 1999) with a hemispheric FoV. These measure global horizontal and diffuse horizontal irradiance (GHI and DHI) in 5 aerosol channels, the difference in GHI and DHI divided by the solar-zenith angle is cosine-corrected to provide calculated direct beam spectral irradiances. The temperature is held near 40 C. The effective FoV is the largest of any of the instruments in this study at ~ 6.5°.

e. Three (3) Precision Solar Radiometers (PSR) that are direct sun pointing spectroradiometers able to measure the spectrum from 300 to 1000 nm with wavelength increment of 0.7 nm. FoV and FWHM is 1.5° and 1.5 to 6 nm respectively. These are manufactured by PMOD/WRC and are temperature controlled.

f. Three (3) direct sun pointing POM-2 sky radiometers instruments from Prede Co., Ltd. The instruments have a FoV of 1° and FWHM equal with 3 nm (UV), 10 nm (VIS) and 20 nm(IR).

g. Four (4) Solar Spectral Irradiance Meters (SSIM) from Cofovo Energy Inc. The instruments measure AOD in 6 wavelengths with FoV equal with 2° and FWHM equal with 5 nm.

h. One (1) Microtops hand-held aerosol sun photometer from Solar light Co. the instrument is a hand help device measuring at 5 wavelengths between 340 nm and 936 nm with FoV and FWHM equal with 2.5° and 10 nm respectively.

Historically, instrument comparisons have consisted of bringing a number of instruments together to a single location for a period of several days to several weeks (e.g., Schmid et al., 1999). These types of comparisons are essential to moving forward the frontiers of instrument and metrology science. However, there may be little or no relation between the results of these intensive comparisons and the results from the same instruments when placed in an operational network setting. The comparison that is reported here provides insight into the quality of data output by instruments when cared for following operational protocols, designed by the various data centers, responsible for the routine handling of the measurements. Therefore, the results of this comparison should provide an understanding of both the comparability between different networks and the overall data quality of participating networks. However, in addition to this comparison's results, homogeneity related conclusions for different Networks are linked with the action of each Network towards standardization of calibration, instrumentation and towards the use of standard operational procedures (SOPs) including data quality control



and quality assurance protocols. Given the differences in instrumentation characteristics, calibration strategies (Walker et al.,
1987) and processing algorithms used by different networks, the effective equivalence of AOD observations needs to be
estimated through Intensive Observation Periods (Schmid et al., 1999) or extensive field comparisons (McArthur et al.,
2003; Mitchell and Forgan, 2003) of co-located instruments representing different networks.

Table 1: Details of sun photometers used during the 4th FRC AOD intercomparison campaign.

| Instrument type | Measuring wavelengths (nm) | Field-of-view (deg) | FWHM* (nm) | Measurement principle |
|---|---|---|---|---|
| PFR-N | 368, 412, 500, 863 | 2.5 | 3.8-5.4 | Sun pointing on tracker |
| CIMEL | 340, 380, 440, 500, 675, 870, 1020, 1640 | 1.2 | 2,4 (340, 380nm) 10 (rest of wavelengths) | Sun pointing on tracker |
| MFRSR | 415, 500, 615,673, 870, 940 | ~6.5 | 10 | Diffuse and global using shadow-band |
| POM-2 | 315, 340, 380, 400, 500, 675, 870, 940, 1020, 1627, 2200 | 1 | 3 (UV), 10 (VIS) up to 20 (IR) | Sun pointing on tracker |
| PSR | 300-1000, step 0.7 | 1.5 | 1.5-6 | Sun pointing on tracker |
| SP02 | 368, 412, 502, 675, 778, 812, 862 | 5 | 5 | Sun pointing on tracker |
| SSIM | 6 filters, spectral AOD retrieval | 2 | 5 | Sun pointing on tracker |
| Microtops | 340, 440, 500, 870, 936 | 2.5 | 10 | Hand held – tripod |

*FWHM = Full width at half maximum

### 2.3 AOD retrieval

AOD is defined as the negative natural logarithm of transmission, normalized to the vertical path length, m = 1 through the
atmosphere, its error becomes proportional to the relative error in calibration and inversely proportional to the length m of a
slant path.  The current GAW specification (WMO, 2005) calls for an AOD uncertainty of 0.005±0.01/m thus requiring a
calibration uncertainty of 1%. This specification is similar to the uncertainty required for satellite AOD retrievals of 0.015
over land and of 0.010 over the ocean in order to make a meaningful statement concerning the aerosol climate effect (Chylek
et al., 2003).



Measurements of solar irradiance were nominally taken each full minute by the participant's data acquisition systems, typically yielding 500 observations per cloudless day. Actual sampling/averaging rates ranged from 15 seconds to 1 minute depending on the instrument. Simultaneous measurements were defined in a timing window of 30 seconds before and after

each full CET minute. The raw measurements were evaluated by each participant according to their preferred algorithms, including cloud-screening, and submitted for comparison.

The set of measurements covered wavelengths between 340 nm and 2200 nm. Channels at 368±3nm, 412±3nm, 500±3nm, 865±5nm were defined as the AOD intercomparison wavelengths. The number of instruments that submitted AOD retrievals for each of those wavelengths is summarized in Table 2:


Table 2: Number of instrument submitting AOD data for each wavelength during FRC-IV.

| Wavelength | Number of instruments |
|---|---|
| 368±3nm | 17 |
| 412±3nm | 21 |
| 500±3nm | 29 |
| 865±5nm | 29 |

Ångström exponents were derived from optical depths at 500 and 865 nm (29 instruments). Values of atmospheric pressure, precipitable water, relative humidity and temperature readings, were made available to all participants by the MeteoSwiss

weather station located at PMOD/WRC, at a 10 minute resolution. Total ozone column content measured with a double Brewer spectroradiometer at PMOD/WRC, was available as well. This common auxiliary database was available to all participants in order to avoid AOD related discrepancies introduced by uncertainties linked with the abovementioned parameters.

Several of the participating radiometers were calibrated at various sites within a few months prior to FRC-IV. Their

performance during this comparison can be used to estimate the homogeneity of AOD observations across weather services, networks or individual measuring sites. For more details about the instrument acronyms, their participation in national or international aerosol networks and their basic calibration technique see table s1 in the supplementary document.

Each of the instruments that have participated in the campaign have been calibrated using techniques that are quite well documented in various publications describing the instrument/network calibrations. More specific:

PFR Instruments: The procedure for the calibration of the reference triad is described in Kazadzis et al., 2017. Two of other PFRs have been calibrated through comparison with the triad in June 2014 and September 2015, respectively. Two PFRs have been calibrated using the Langley technique for a 6 month period at Izana, Tenerife (February-August, 2015) and one



using the same technique at Mauna Loa, USA. Finally one PFR have been calibrated through Langley related measurements at Davos, Switzerland.

CIMEL instruments: The Cimel sun photometers (#627 and #917) were calibrated by the Langley plot method at the high altitude station Izaña following the AERONET protocols for master instruments (Holben et al., 1998), just before the campaign (August 2015). $V_o$ values were calculated as the average of five different Langley calibrations in June, 2015 (mean AOD at 500 nm < 0.016 during these days), following the criteria based on the coefficient of variation (CV) determined in Holben et al. (1998). This criteria requires CV < 0.5% for VIS and IR spectral bands and 1% for UV wavelengths. The

permanent Cimel at Davos (#354) was calibrated by comparison with an AERONET master instrument in June 2015, following the AERONET standard procedure for field instruments.

POM instruments: The calibration for two POM instruments is obtained every month through the improved Langley technique (Campanelli et al. 2004) at the respective stations. The method, is based on the processing of the almucantar measurements, has proven to be accurate and does not require a stable aerosol optical thickness as the normal Langley

extrapolation. One instrument was calibrated by an outdoor comparison to the Japanese Meteorological Agency (JMA) reference POM-02 (May 2015 - August 2015). JMA reference POM-02 was calibrated by the Langley technique at Mauna Loa, USA.

MFRSR instruments: SURFRAD network MFRSRs are calibrated on site using a robust estimate for $V_o$s from Langleys based on at least one month or more of data in representative conditions (Augustine et al., 2003). MFR_US_2 and

MFR_US_3 were calibrated using only the data from the FRC-IV; MFR_US_1 and MFR_DE_1 also used the data from FRC-IV to calibrate following slightly different modified procedures to determine Vos because of the short duration of the campaign.

SPO2 instruments: The Australian Bureau of Meteorology SPO2s were removed from a high frequency clear sun Australian (Longreach) station where they were calibrated in-situ for 2 years using the methods described in Mitchell, Forgan and

Campbell, 2017, prior to participating in the FRC.

PSR instruments: PSR instruments were absolutely calibrated at the PMOD/WRC laboratory during the campaign. In order to retrieve the AOD, an absolute extraterrestrial solar spectrum is used.

Microtops instrument: The instrument was calibrated by direct comparison with a calibrated CIMEL/AERONET instrument from June to August, 2015.

COFOVO instruments: The four instruments are calibrated through direct comparison with the National Renewable Energy Laboratory, USA secondary reference spectoradiometer (Tatsiankou et al., 2013). AOD is retrieved by matching absolute irradiances at the six measuring wavelengths with a radiative transfer model.

During the intercomparison, AOD data delivered by the operators of the participating radiometers were evaluated using a common comparison software. The comparison was based on AOD results only, as each operator/group used their own

algorithm normally used for standard radiometer operation. The comparison principles were based on the recommendations



formulated during the WMO experts workshop on "Global surface network for long term observations of column aerosol optical properties", held in 2004 in Davos (WMO, 2005), which called for:

- At least 1000 data points (1-minute data) with AOD at 500 nm between 0.04 and 0.20
- A minimum duration of 5 days
- Traceability requiring 95% uncertainty within ±0.005 + 0.01/m optical depths

During FRC-IV, weather conditions allowed over 1000 measurements to be made for most instruments on 5 days, allowing the above-mentioned recommendations to be fulfilled.

## 3 Results

### 3.1 AOD differences

This comparison is based on AOD values provided by the individual instrument operators against the triad. Figure 2 shows an example of this comparison including various instruments separated in different instrument types, against the PFR triad on a diurnal plot. The majority of the PFRs showed the best performance with absolute AOD differences from the triad
ranging in all cases and wavelengths from zero to 0.01. As the measured wavelength increases, the errors are minimized reaching performance errors close to zero except for some overestimated outliers for PFR_SE_N35, which were caused by non-synchronous measurements (timing) for particular periods. Results for the three CIMEL (CIM) instruments are almost identical to those of the PFR at 500 and 862 nm (Fig. 2a), while a slight underestimation in the order of 0.01 and 0.005 at the shorter wavelengths 368 and 412 nm, respectively was found. It has to be noted that CIMEL AOD at 412 and 368 nm have
been linearly interpolated using the CIMEL AOD at 340, 380 and 440 nm and the AEs derived from these three wavelengths. So part of the difference can be explained by the interpolation related uncertainties. POM sky radiometers do not measure AOD at 368 and 412 nm. However, comparable results to the CIM and PFR at 862 nm was retrieved, with a slight underestimation, well within the WMO limits, at 500 nm (Fig. 2b) which was not related to the air mass. This proved the high level of the quality of reference instruments belonging to the GAW-PFR, AERONET and SKYNET networks. The
two SPOs, which are similar instruments to the PFRs but with a wider field-of-view and with no temperature controller, showed good agreement compared to the triad. SPO_AU_1, showed excellent median differences (Fig. 2c). For the SPO_US_1, one of the five days of measurements at 500 nm and one of the five days at 862 nm were overestimating, with excellent agreement other days and excellent agreement on all days at 500 nm. The overestimates were likely the result of the four FoVs of the SPO not being optimally aligned. During the shipment of the SPO2_US_1 to Davos the diopter was
damaged. It was manual adjusted to its position during the FRC without the benefit of a detailed alignment process that is usually followed to minimize the misalignment of the four independent barrels of the sunphotometer. At 368 nm, small SPO_AU_1 calibration related AOD differences were observed compared to the triad. The four MFR instruments showed



good agreement for the medians compared to the PFR triad, however, they exhibit larger scatter than the sun-pointing instruments resulting in a lower precision. McArthur et al. (2003) had previously reported that the MFR-derived AOD does

not quite meet the accuracy of the sun-pointing instruments under clean atmospheric conditions. MFR_DE showed an AOD overestimation in various instances that gave results that are outside the WMO defined AOD limits (Fig. 2d). This small overestimation of the MFR_DE instrument compared to the PFR Triad could be due to uncertainties introduced while correcting for their angular response, the calibration procedure, or incomplete blocking of the diffuser by the shadow-band. The MFRSRs that are part of the SURFRAD network (MFR_US2 and MFR_US3) give a median AOD at 500-nm that is in

very good agreement with the PFR triad and as good or better than some of the other sun-pointing instruments, e.g., CIMEL and POM; these two slightly underestimate the AOD at 865 nm, but are within the WMO defined limits.   Again, these two MFRs' medians are comparable to the better sun-pointing instruments, but give larger scatter.   These two MFRs are representative of the SURFRAD network that follow network protocols for calibration and alignment and frequent characterizations of the spectral and angular responses (Augustine et al., 2003, Michalsky et al., 2001). Again, this highlights

the high level of the quality of instruments that represent larger networks (GAW-PFR, AERONET, SKYNET, and SURFRAD networks).

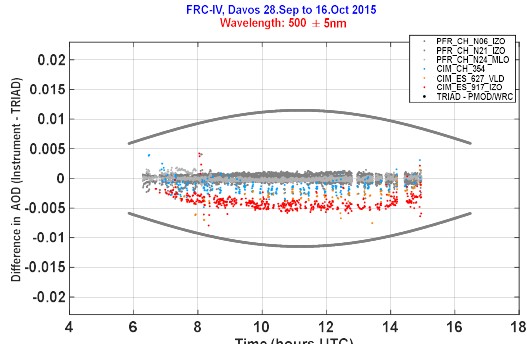 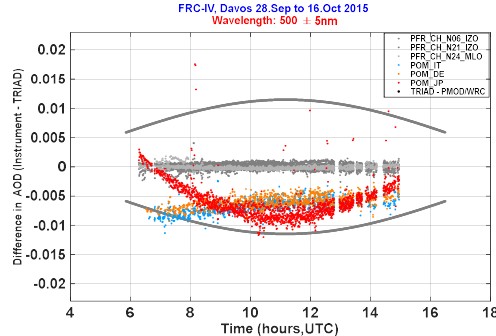

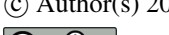


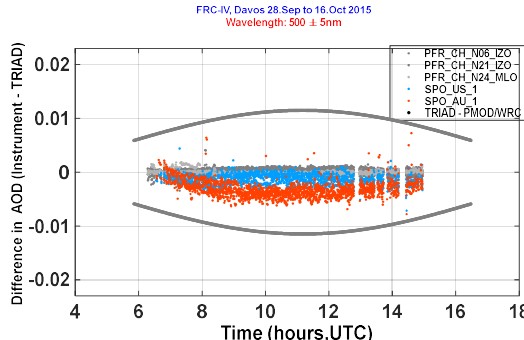
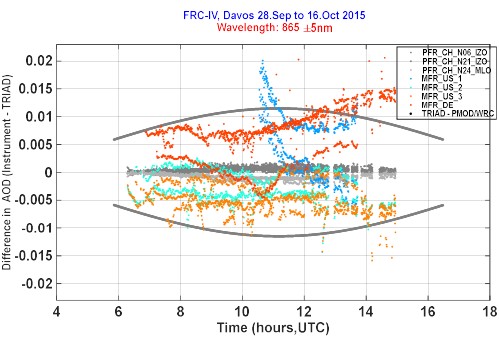

Figure 2. Comparison of the triad (grey points), with the: CIMEL instruments (upper left (a), 500±5 nm), POM instruments (upper right (b), 500±5 nm), SPO instruments (lower left (c), 500±5 nm), and with the MFR instruments (lower right (d), 862±5 nm). Different colors represent different instruments for all the five comparison days and grey lines represent the WMO AOD limits.

Figure 3 shows the comparison results in terms of absolute difference between the triad and the 9 individual PFR instruments, 3 CIM (AERONET), 3 POM (SKYNET), 2 SPO, 4 MFR, 3 PSR, 4 SIM and 1 MIC instrument. The box-plots represent the range between the 10[th] and 90[th] percentiles with the in-box dot showing the median, and the upper and lower whiskers showing the maximum and minimum error value information that is within 1.5 times the inter-quartile range of the box edges. The figure shows the good agreement among most of the instruments compared to the reference triad. WMO limits cannot be shown in Figure 3 as they are air mass dependent. However, for FRC-IV these limits were between 0.006 and 0.012 for low solar elevations and local noon, respectively.

PFR AOD comparisons showed that median differences were well within ±0.005 with the 10[th] to 90[th] percentiles also well within ±0.01 AOD limits, at all wavelengths. Similar results were found for CIMEL AODs at 500 and 862 nm. POM AOD medians showed a small underestimation of about 0.005 at 500 nm and very good agreement at 865nm. The medians of the MFRs AODs were within 0.01 AOD except for the MFR_DE_1 at 500 nm. The three PSR instruments are the only ones that provide high spectral resolution AOD measurements, and the comparisons highlighted the accuracy of the medians at longer wavelengths (500 and 862 nm) with a tendency of overestimated outliers, and a 0 to 0.02 discrepancy between the PSRs at shorter wavelengths. Overall, better results were demonstrated by PSR_006. SIM instruments showed an excellent agreement at 500 and 865 nm, an overestimation from 0.01 to 0.03 and higher scatter than the other instruments. However, based on the instrument retrieval methodology (use of a radiative transfer model with direct irradiances as inputs, in order to calculate AOD), the results can be considered as very good. Finally, the hand-held Microtops instrument overestimated at the two shorter wavelengths, while the scatter of the differences was 0.01 to 0.04 for the 10[th] to 90[th] percentiles. The blue lines in figure three are defined at the -0.09 and 0.09 AOD limits. This is an average of the air mass related WMO limit that ranged





from 0.06 to 0.12 for the campaign period. CIMEL AOD at 368 and 412 nm has been interpolated using the CIMEL AOD at 340, 380 and 440 nm, and the Ångström exponents derived from these three wavelengths.

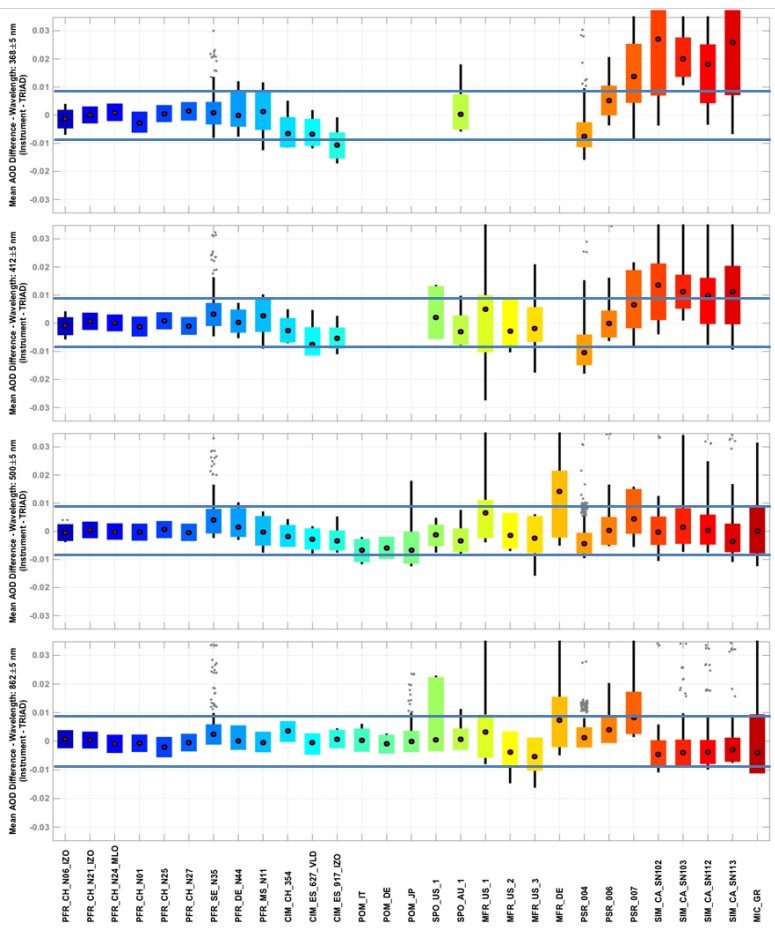


**Figure 3a-d:** AOD comparison results at 368±3nm (a), 412±3nm (b), 500±3nm (c), 865±5nm (d). The black dots represent the median of the difference of each instrument from the mean of the triad at each wavelength over the five FRC-IV selected days. The boxes represent the 10[th] and 90[th] percentiles while the black lines represent the minimum and maximum values of the distribution excluding the outliers. Outliers (gray dots) represent values that are outside the 10[th] and 90[th] percentiles by four times the width of the distribution at a 10% level. CIMEL AOD at 368 and 412 nm has been interpolated using the CIMEL AOD at 340, 380 and 440 nm, and the Ångström exponents derived from these three wavelengths. Box colors are only used to differentiate between instruments. Blue lines represent the ±0.09 limits.




Overall, the FRC-IV intercomparison results are comparable with the results found by Mitchell and Forgan (2003), Mitchell
et al. (2017) and Kim et al. (2008) under low aerosol loading conditions. The magnitude of the instrument's discrepancy
could be partly due to the inherently different spectral responses and detector fields-of-view of each instrument under
varying aerosol loadings (Kim et al., 2005). The above results indicate that the pointing instruments provide data of
comparable quality. On an observation-by-observation basis, the direct-pointing instruments appear to maintain a difference
of lower than 0.01 at nearly all wavelengths in clear stable conditions, equal or lower than the AOD uncertainty. It is
estimated that advances in the following aspects may improve (see Section 3.3) agreement at the 0.005 level: i) instrument
pointing, ii) better determination of the effects of Rayleigh scattering, ozone, and other absorbers on the calculation of AOD,
and iii) better instrument characterization, especially calibration of the radiometers. Significant improvements in AOD
precision and instrument accuracy were obtained upon application of cloud-screening algorithms.

Concerning additional statistics, we have used Taylor (Taylor, 2001) diagrams in order to evaluate the performance of all
instruments at the four measuring wavelengths (Figure 4). Correlation coefficients (CC) among the triad and all other
instruments were better than 0.9 for all instruments and wavelengths, with the exception of three instruments, only at 865
nm. In the case of the CIMEL, PFR and POM, CCs were higher than 0.98 in all cases. The normalized standard deviation in
Fig. 4 describes the instrument measured AOD variability compared to that of the reference (triad). Most of these ratios were
well within the 0.8 to 1 area, with the exception of a single PFR instrument, that provided data for only one comparison day.

Overall, statistics at 368, 412 and 500 nm showed an excellent agreement for all instruments, while at 865 nm the instrument
scatter within the Taylor diagram space is higher. However, the agreement can still be considered quite good, as seen when
examining Figure 4.

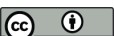



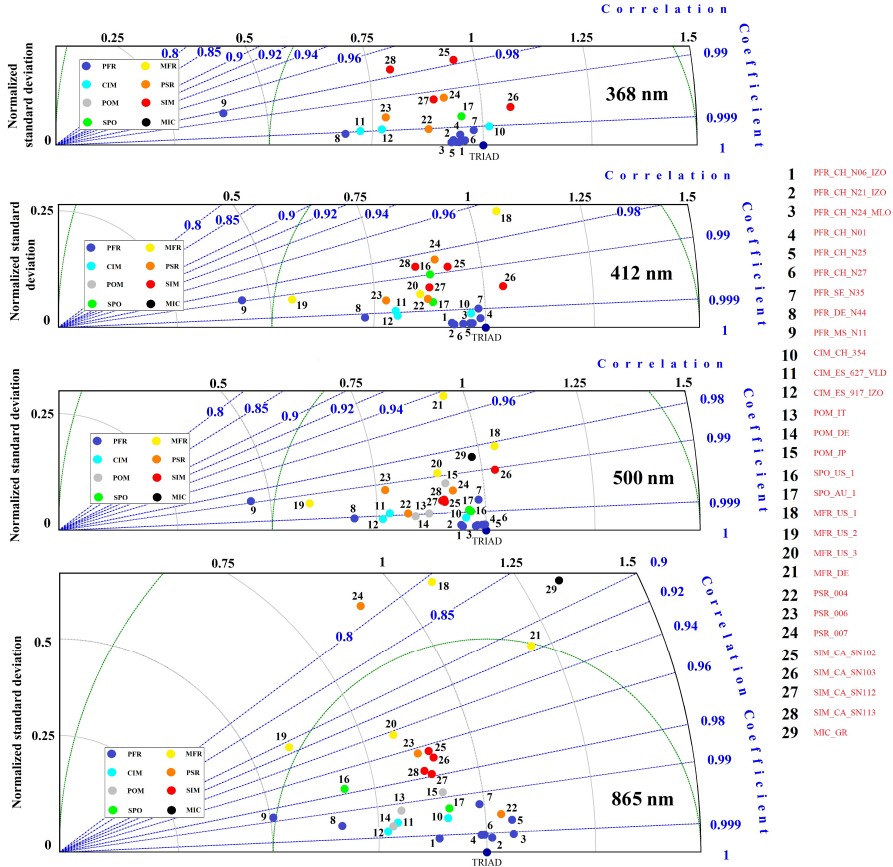

Figure 4: Taylor diagrams at the four measuring wavelengths.


Figure 5 presents the percentage of instruments that lie within the WMO AOD uncertainty criterion. The wavelengths with the lower percentage of instruments within the defined criterion are the nominal 368 and 412 nm channels, while the majority of instruments measure within the defined criterion for the nominal 500 and 865 nm channels (see Table 2). When considering 95% of measurements, the best results correspond to the 500 nm wavelength followed by the 867 nm

wavelength. A main finding is that the lower the wavelength, the lower is the reliability, accompanied by the lower percentage of participating/supporting instruments. For a lower percentage of measurements (horizontal axis) the 865 nm wavelength reaches 100% of participating instruments, which decreases to 83% at 95% of data within the WMO limits. The shortest studied wavelength (368 nm) showed that 12 out of 17 instruments were within the WMO criterion while the remaining five had less than 70% of the comparison data among the WMO limits.



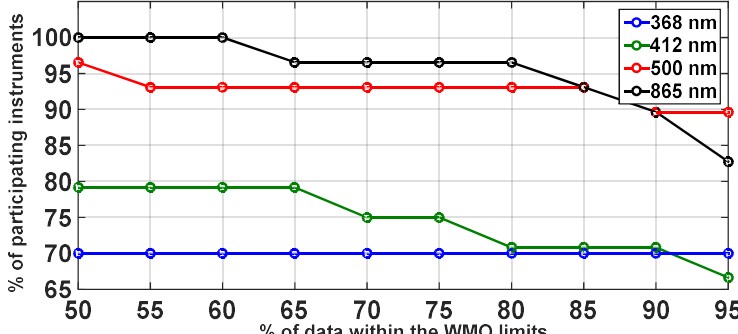


**Figure 5:** Percentage of instruments that lie within the WMO criterion (0.005 + 0.01/m optical depths). The horizontal axis shows the different percentages of measurements within the criterion ending at 95%, which is the U95 WMO limit.

The difference in the AE between all participant instruments and the triad is shown in Figure 6. We have used only the 500/865 nm channels to calculate the AEs in order to have the same calculation principles for all instruments.

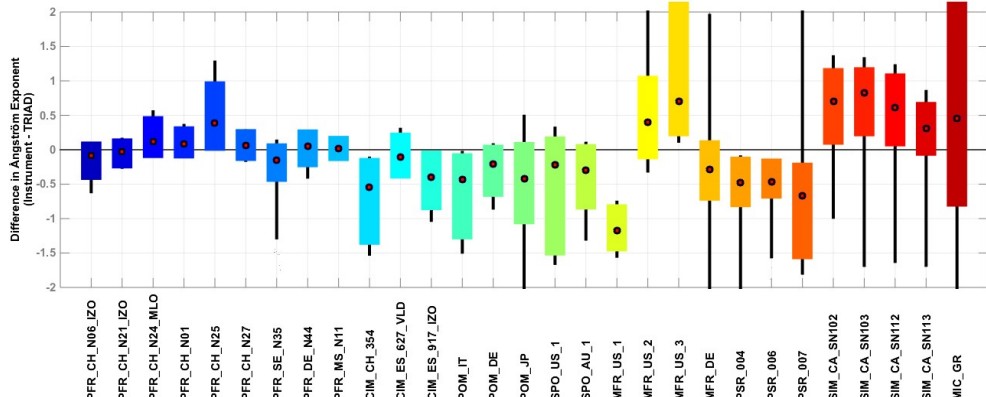


**Figure 6:** Difference in the Ångström exponent between each instrument and the WORCC triad. The boxes represent the 10th and 90th percentiles while the black lines represent the minimum and maximum values of the distribution excluding the outliers. Outliers (gray dots) are considered to be values outside the 10th and 90th percentiles by four times the width of the distribution at a 10% level. Box colors are only used to differentiate between instruments.

Under low aerosol conditions, a small relative bias in the AOD determination at 500 and 865 nm can theoretically lead to large deviations in the calculated AEs. As an example of AODs of about 0.05 and 0.02 at 500 and 865 nm, respectively, AOD differences of 0.01 and 0.005, respectively, can lead to AE differences up to ~1. This was observed during FRC-IV, and Figure 6 shows that for such low AOD conditions, AEs can differ substantially. Most of the AE instruments differ from the one triad calculated AE by the triad, by less than 0.5 (median difference) but the 10th to 90th percentiles are about 0.5 for





the PFRs and close to 1 for all other instruments with the exception of the Microtops instrument retrievals that showed a very large variability in AE difference.

## 3.2 Cloud flagging

The FRC campaign was a unique opportunity to compare the different cloud-screening algorithms used by each instrument/group.  McArthur et al. (2003), have reported on instrument/network related cloud flagging differences using a

three month campaign measurements. The use of such algorithms can lead to significant differences while the selection of threshold values to filter out the retrievals could lead to large deviations. For our comparison, we have used one of each of the main types of instruments and compared the number of available retrievals (PFR, POM, SPO, MFR and CIMEL instruments). More specifically, we have chosen to examine the instruments of each type with the larger dataset on these five days.

The cloud detection algorithm used for the above-mentioned instruments can be summarized as:

CIMEL: The AERONET operational cloud-screening algorithm, described by Smirnov et al. (2000; 2004), was used. It consists of temporal filtering in several steps, from minute (triplet stability, with AOD variation <0.02) to hour and diurnal checks, that impose restrictions on the AOD second derivative with time as well as the standard deviation of AOD within the day.

PFR:  Three different criteria are used (Wehrli, 2008): a. The instrument signal derivative with respect to air mass is always negative (Harrison et al., 1994). For cases of air masses < 2 where a cloud influence on the noon-side of a perturbation cannot be easily detected, a comparison of the derivative with the estimate of the clear Rayleigh atmosphere is performed and data are flagged as cloudy if the rate of change is twice as much (objective method); b. A test for optically 'thick' clouds with AOD >2 is performed; c. Use of the Smirnov et al. (2000) triplet measurement by calculating AOD and looking at the

signal variability for three consecutive minutes (triplet method).

POM: The Smirnov et al. (2000) algorithm, was implemented in the SUNRAD code which was used for the POM instruments (Estellés et al., 2012) with two main differences related to instrument characteristics. First, the minimum signal threshold is set to 5.0E-7 A. Second, triplets are built *a posteriori* with 1-minute instead of 30-second data, as used in the CIMEL. A further check was introduced in the current version of the processing software, which is consistent in removing

isolated AOD data points, namely: a given AOD point will be flagged if the previous and next AOD values were already flagged in the standard application of the cloud screening algorithm.

SPO: The selection of valid AOD values for a sample time is made up of three components with the first two related to measured signals and the last one based on the estimated AODs plus cloud/shade value. For each time sample, if the 868 nm (10 nm FWHM) signal was below a standard threshold, all wavelength channels are cloud-flagged or not oriented to the Sun.

Secondly, if the maximum signal of all wavelength signals for a sample time was less than 10 times the resolution of the data acquisition system, all wavelength signals for the sample time are flagged as being cloudy or shaded.  Lastly, for each wavelength and for all remaining signals, the AOD is derived and the Alexandrov et al. (2004) algorithm with a time span of





15 sequential samples is used to examine each wavelength's AOD time series; if AOD at any wavelength is rejected by the algorithm for a sample time, the AOD is deemed to be affected by clouds at all wavelengths.

MFR: The technique used for MFRs is described in Michalsky et al. (2010). A coarse filter is used on ten minutes of data that examines differences first from 20-sec sample to 20-sec sample and then over the entire 10-min interval. This is followed by a second similar filter, but using allowances of variability that is scaled to the approximate value of the AOD. If the 10-min span passes both tests, the test is repeated after advancing one 20-sec sample. Duplicate points from processing all of the data are discarded.

We have used the tool developed by Heberle et al. (2015) to visualize the coincidence of the instrument data sets that provided 1-minute AOD (SPO, MFR, PFR and POM) by plotting Venn Diagrams (Figure 7). CIMEL instruments were not included due to the lower AOD-measuring temporal resolution. All instruments only detected cloudless conditions during 25% of the common measurements. The SPO seems to have the most values that do not appear in common with other instruments (4.9% solo, and 18% in common with only one other instrument) and the POM the least (0.1% and 0.8%,

respectively). When considering measurements derived as cloudless from at least three out of four instruments, the SPO has the largest number of coincident measurements (69.9%) followed by the PFR (69.2%), MFR (59.9%) and POM (36.3%). The POM has the smallest dataset, only retrieving AOD from 40% of all possible (at least one instrument provided cloudless AODs) measurements.


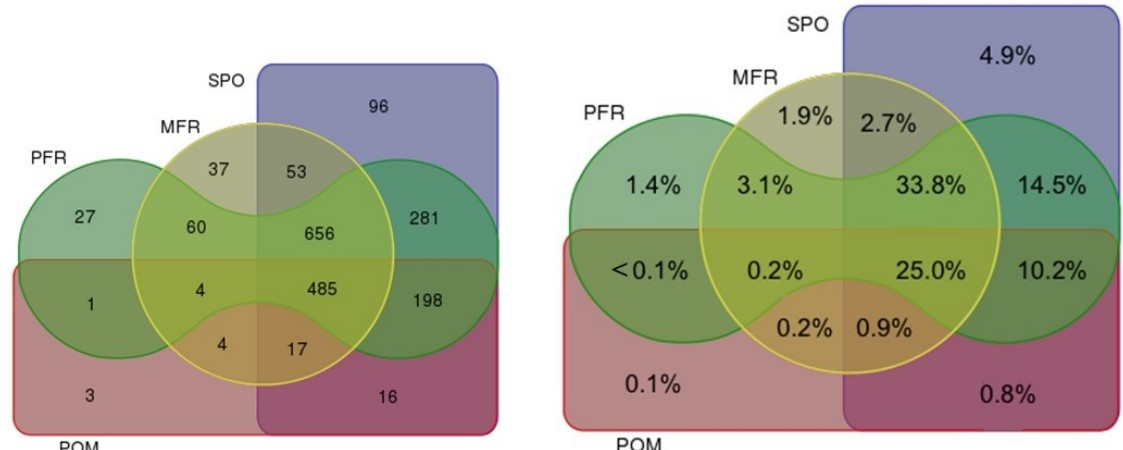

**Figure 7:** Venn diagram of quality controlled, clear sky data sets of SPO, MFR, PFR, and POM data for four cloudless days. Left panel: number of measurements, right panel: percentage of measurements.




In order to investigate measurements when only one instrument provides cloud free minute measurements while all other instruments are marked as cloud-flagged (as an example in Figure 7: the SPO has 96 cases/minutes out of a possible 1944 comparison data/minutes), we calculated an artificial AOD time series. This was constructed by spline-interpolating the mean AOD of all the remaining (three) instruments (excluding the CIMEL that has a lower temporal measurement frequency

than the rest of the instruments), at the time intervals where the fourth instrument (SPO in this example) provides cloud free data. It was found that the mean AOD at 500 nm ($AOD_{500}$) and the SPO retrieval difference is 20.5%. In this example, on the one hand a 20.5% increase of AOD over one or a few minutes could be considered as a reason of rejection (cloud-flagging) for all other algorithms except that of the SPO. However, a difference of 0.006 in optical depth could be considered as a limit on trying to separate aerosol and very thin cloud attenuation.

In Table 3, we have calculated the score for each instrument, dividing the number of available retrievals by a total of 1944 possible (at least one of the instruments has provided an AOD cloud free minute value) comparison cases. For CIMEL values, where the measurements are not every minute, we used raw data to count all the recordings, and divide the number of cloud-screened data, so it is not directly comparable with other instruments. The POM instruments obtained the lowest score in the cloud screening application, mainly caused by the stringent isolation-check added to the adapted Smirnov et al. (2004

algorithm.

Table 3. Percentage of available cloud-screened AOD data values out of all possible measurements (minutes).

| Instrument type | Score % |
|---|---|
| PFR | 88.4 |
| POM | 39.7 |
| SPO | 89.1 |
| MFR | 70.9 |
| CIMEL | 82.1* |

\* taking into account the CIMEL measurement frequency





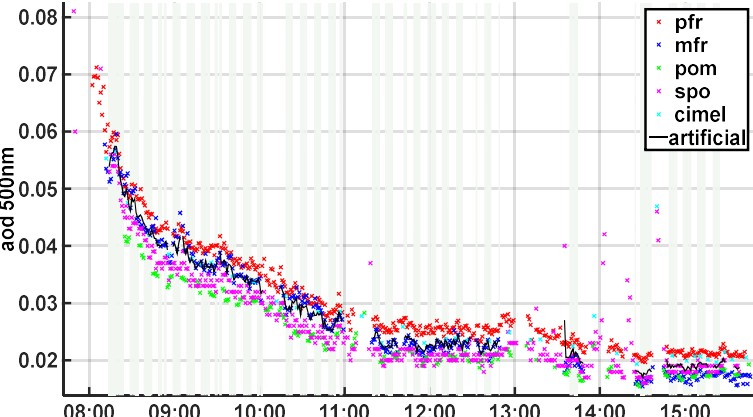

Figure 8: One-minute AOD data on October 1, 2015. Different colors represent the AODs, submitted as cloud free data. The black line is the mean AOD from the PFR, MFR and SPO for data points when all three instruments provided data. The gray vertical lines represent periods where the PFR, MFR and SPO provided data but the POM characterized them as "cloudy".

Figure 8 shows AOD measurements at 500 nm for all instruments that were tested for their cloud-flagging algorithms during

one single day. As seen in Table 3, the POM instrument seems to cloud-flag various minutes/measurements, while all other instruments/algorithms do not. Such instances are shown in Figure 8 as gray areas, and represent periods when all PFR, SPO and MFR instruments provide AOD (thus they do not "detect" any clouds) while the POM does not provide an AOD. Despite the small instrument-to-instrument differences, the evolution of the AOD during particular periods (gray areas), also described by the mean or artificial AOD, cannot be considered as periods that are affected by clouds. Thus, the POM

algorithm is probably too strict compared to the others. In addition, sporadic SPO related high AOD values after 14:00 (at times when no other instrument provides cloudless data) show that during these conditions, the SPO cloud-flagging algorithm was more imprecise.

### 3.3 AOD retrieval differences

For the present intercomparison, no common procedures were used for the removal of gas phase constituents or Rayleigh

scattering; cloud screening, solar position, timing, and calibration methodology were at the discretion of the network operators. Datasets from each sun photometer network were corrected for these factors independently. Figure 9 identifies some of the possible discrepancies that may result when considering $NO_2$, ozone, Rayleigh scattering, other trace gases and $H_2O$ in the atmosphere (Thome et al., 1992) at 500 and 870 nm during the 4th FRC.

One reference day (September 30, 2015) was chosen for this comparison exercise. The slant optical thicknesses of various

trace gases and Rayleigh scattering were obtained from CIMEL, PFR, POM and SPO instruments and individually

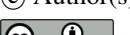



compared. Furthermore, the respective algorithms for the calculation of the solar zenith angle and air mass at any given time (as provided by the responsible scientists of each instrument) were employed. $NO_2$ absorption was considered only for POM (fixed vertical column density of 0.218 DU for mid-latitude summer; method and cross-sections from Gueymard, 1995b and Gueymard, 2001) and CIMEL instruments (SCIAMACHY monthly climatology; cross-sections from Burrows et al., 1998)

and only for AOD retrieval at 500 nm wavelength. Ozone absorption was taken into account by all instruments at 500 nm, but was not accounted for by the CIMEL at 870 nm. Different ozone amounts (measured value of 314 DU for PFR and SPO; fixed value of 300 DU for POM; OMI climatology for CIMEL) and cross-sections (Gueymard, 1995 for PFR; Gueymard, 1995b and Gueymard, 2001 for POM; Burrows et al., 1999 for CIMEL; custom set of ozone coefficients for SPO) were adopted. The Rayleigh scattering coefficients by Bodhaine (1999) are used by all networks except SPO, which use those by

Bucholtz (1995). Pressure was measured (845.7 hPa) by the PFR control-box, while it was fixed and corrected for altitude (z) for the POM (840 hPa) using the formula:

$$p = 1013.25*\exp(-0.0001184*z) \qquad (1)$$

Finally, water vapour is only taken into account by POM instruments using a fixed value for the summer season and additionally corrected for altitude using the following formula based on Gueymard, 1995 data.

$$w = 2.9816*\exp(-0.552*z) \qquad (2)$$

Where w is the precipitable water in cm, and z is the altitude in km. The method for deriving the corresponding $H_2O$ optical depth is also adopted from Gueymard, 2001. Results of this comparison exercise are shown in Figure 9.

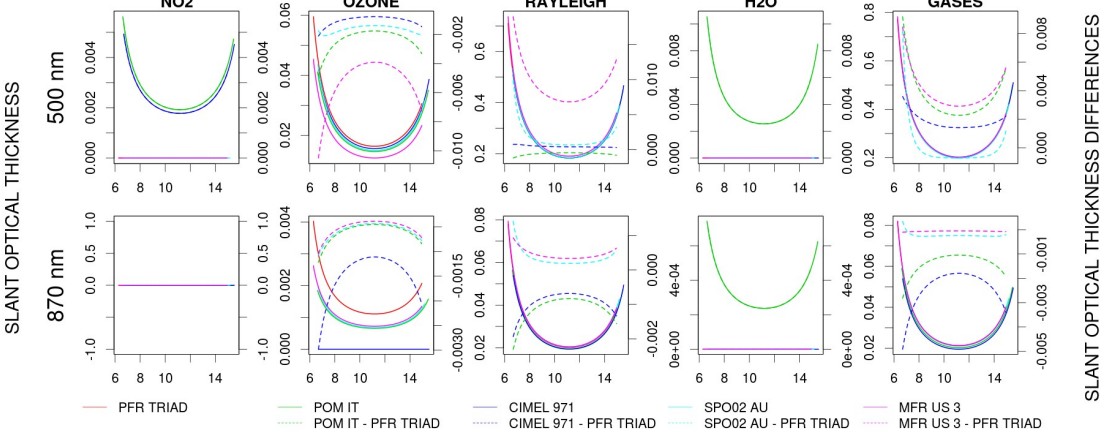





Figure 9. Slant column optical thickness (right axis – thick lines) and optical depth differences compared with the triad (left
axis dashed lines), at 500 (upper panel) and 870 nm (lower panel).

The analyzed factors result in discrepancies of comparable magnitude at a wavelength of 500 nm, but also illustrate a slightly larger effect due to differences in the corrections for Rayleigh scattering and water vapor. At 870 nm, the larger discrepancies can be ascribed to different parametrizations of ozone absorption and Rayleigh scattering. For the case of the

MFR instrument the effective wavelength of the "500-nm" filter is about 495.8 nm, which explains the higher Rayleigh optical thickness and the lower ozone absorption related one. The deviations between algorithms can be of either sign and can partially compensate each other in AOD calculations. Finally, $NO_2$ related differences were 0.002 to 0.004 at 500 nm, at a location (Davos) with very low $NO_2$ columnar concentrations. The error in the (vertical) AOD resulting from differences between the algorithms (obtained by dividing the differences in the slant optical thicknesses by the air mass factor) did not

exceed 0.005 for the selected day. This value is far below the traceability threshold and can thus be considered negligible.

## 4. Summary and conclusions

Results from the FRC-IV have been presented in this study. Based on the number of instruments and also the participation of reference sun photometers/instruments from various global AOD networks, the campaign could be considered as a successful experiment in assessing the current status of AOD measurement accuracy and precision. The WMO

recommendations for AOD comparisons have been adopted for the present campaign and the WORCC PFR triad has been used as a reference.

The absolute differences of all instruments compared to the reference triad have been reported and are based on the WMO criterion defined as: "95% of the measured data has to be within 0.005±0.001/m". At least 24 out of 29 instruments achieved this goal at 500 and 865 nm, while 12 out of 17 and 13 out of 21 achieved this at 368 and 412 nm, respectively.

The statistics from the Taylor diagram analysis revealed the overall accuracy and homogeneity of the instruments. In particular, the majority of instruments gave CCs >0.98 and a normalized standard deviation in the range 0.75 - 1 as compared to the triad, at all wavelengths. The similarity of results and the high accuracy of the PFR, CIM and POM instruments demonstrates a promising framework to achieve network homogeneity in the near-future, concerning the AOD measurements. The PSR spectroradiometers, SIM and SPO filter radiometers also had CCs over 0.96 under all conditions.

Ångström exponent calculations using a pair (500 nm/865 nm) of wavelengths showed relatively large differences among different instruments. This was largely related to the sensitivity of this parameter at low AOD conditions. AOD differences of about 0.01 at 500 nm that can be easily related to the instrument calibration uncertainties can considerably affect such calculations during low AOD conditions. Hence, this campaign reaffirms that for cases of mean $AOD_{500} < 0.1$, the calculation of AE becomes highly uncertain.



Investigating the sources of differences among different instruments, we compared all parameters included in the AOD retrieval algorithm as provided by the different participating institutes. All individual differences (Rayleigh, $NO_2$, ozone, water vapor related optical depths and air mass calculations) amounted to less than 0.01 in AOD at 500 and 865 nm.

Different cloud-flagging algorithms can affect the AOD datasets as different instruments/networks use different techniques. During a day with sporadic appearances of high and mid-level clouds (which was deliberately chosen as a "difficult" task for

such algorithms), results from different cloud-flagging algorithms limited the AOD comparison datasets between two instruments from 40% to 90%, depending on the pair of instruments used, compared to the maximum number of cloudless data points calculated by all instruments. In general, using long term series for determining aerosol climatology at certain locations, too conservative cloud screening could lead to the elimination of high AOD local events, while screening that not eliminates cloud contamination will introduce biases linked mainly with cirrus clouds. Both approaches will have an impact

on aerosol climatology and calculated AOD trends.

In comparison to earlier FRCs (I to III), the latest FRC reported here experienced an increase in both the number of instruments (total of 30) and international participating institutes (12 countries). In addition, analysis at four different wavelengths was performed for the first time. The CIMEL/AERONET, PFR/GAW and POM/SKYNET and SPO participating sun photometers showed very good agreement when compared to older intercomparisons. As AOD from

algorithm differences was quite small, the results of the comparisons of this instrument group are considered to have been very successful as differences are in most cases well within the calibration and overall instrument AOD uncertainties. The rest of the instruments also showed reasonable agreement with few exceptions. MFR instruments experienced additional uncertainties concerning the diffuser based measurements. SIM instruments also performed quite well when considering the radiative transfer based processing algorithm. In addition, spectral AOD retrieving PSR instruments also performed well,

especially at the two higher wavelengths. Finally, Microtops AOD data were in most cases within reasonable agreement with the reference triad but additional technical issues such as the hand-held based sun pointing and the smaller integration time (compared with other instruments) of the direct sun measurement lead to enhanced scatter of the results.

Instrument technical features such as differences in the field-of-view did not play an important role in FRC-IV for the low aerosol load conditions that were encountered. In order to quantify such features and similar issues, intercomparison

campaigns have to be organized in moderate to high AOD conditions when forward scattered radiation and circumsolar radiation can play an important role in instruments with different field-of-view entrance optics.

The results of the FRC-IV, which included a large variety of AOD measuring instrumentation via the participation of reference instruments from AERONET Europe, SKYNET, GAW-PFR, SURFRAD and the Australian aerosol network, could be considered as a starting point for global AOD homogeneity initiatives. The ultimate objective is a unified AOD

product to be used for long term aerosol and radiative forcing studies, case studies involving accurate AOD retrievals, and satellite validation related activities.





**Acknowledgements**

FRC-4 has been organized in the frame of the World Radiation Center - WORCC mandate for
homogenization/harmonization of AOD measurements as defined by WMO-GAW. Authors would like to thank Christian
Thomann for his essential and continuous technical support during the FRC-4[th] campaign.

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
