# Peer review of "Results from the 4th WMO Filter Radiometer Comparison for aerosol optical depth measurements"

_Atmospheric Chemistry and Physics, 2017_

## Referee Comment (RC1)

Review of Kazadzis et al. 'results from the 4th WMO filter radiometer comparison for AOD measurements', submitted for publication in ACP.

The authors describe the results form an intercomparison campaign in Davos, in which the results from AOD measurements in very clear conditions, with very low AOD, are compared. The intercomparison included 30 instruments of different design, from 12 countries. The evaluation of the different instruments and procedures used to measure AOD at different wavelengths, and thus the Angstrom Exponents, contributes to better understanding the results in different AOD networks and eventually to achieve world-wide homogenization of the networks.

The MS is well written and I recommend publication in ACP after minor revision. Below are some comments for the consideration of the authors. One general comment is that the quality of the figures needs improvements: fonts are too small and I cannot see the different data sets from their colours because symbols are very small, certainly in the legends. On screen is even worse than in my print version. This may be the reason why I do not understand all features in the figures.

2, 49 (page, line):      what is meant with 'the least common denominator' in this context? I think it is very common to use AOD in such comparisons.

3, 67:   may be good to define 'air mass'? What is 'relative air mass'?

4, 114:  methods for

5, 126: figure caption: not sure whether WORCC triad is common knowledge: suggest to refer to the text for explanation

6, last para: In the routine handling of instruments in operational networks, also the maintenance and transport and installation are important factors which may influence the results. Are on-site procedures to check the instruments after installation part of the routine? Are traveling standards used and site visits to check on procedures and maintenance? See also what is written on manual adjustment on 10, 285: is this done at the operational sites too?.

7, 184: instrumentation characteristics, calibration strategies and processing algorithms: is this info available for each network, and if so, would it be possible to provide a table where the most recent info can be obtained (may be the networks websites?).

7, 195: Chylek seems to be a rather old and may be a randomly selected reference, I believe that the most recent version of the GCOS requirements is more relevant for satellite measurements (https://public.wmo.int/en/programmes/global-climate-observing-system).

9, 223:  has instead of have?

9, 233:  remove comma after method,

9, 251:  insert comma after USA

10, 267:      This comparison, what does 'this' refer to?;

10, 268:      separated in groups of different instrument types? (add groups of?)

10, 274:      the shorter wavelengths are not shown (unless I miss them, see my general comment on figure quality)

10 279: why does 'this prove the high level of the quality of reference instruments'?

10, 283:       on other days (add 'on')

Figure 2: I cannot discriminate well between the colours but it seems to me that the 3 PFRs mentioned at the top of each legend are the same in each of the 6 plots, as well as the triad. Are these 3 PFRs together the triad? Is that why they are shown each time? And if so, why is the triad shown as a separate item?

15, 365:       the lower the wavelength, the lower the reliability: is there an explanation for this? May be the increased Rayleigh scattering at lower wavelengths? Or molecular absorption?

16, 381:       As an example, for AODs ….

17, 389:       suggest: differences using measurements form a three ….

17, 391:       could lead to large deviations: what do you mean with that? Do you mean that you'd miss the higher AOD cases?

18, 427 and 19, 452:   not every minute? On 6, 157 (bullet c) is mentioned that CIMEL measures within a few seconds. Was CIMEL Used in it's routine operational mode, i.e. every 15 min? Or was it adjusted to continuous measurements to match the other instruments in the intercomparison? May be a few words should be said about his in the text?

Fig. 7, caption: four cloudless days? How does that compare with 5, 129?

19, 443:       what do you mean with 'artificial AOD time series' (also in legend of Fig. 8)? In the caption of Fig. 8 you call them mean AOD, which looking at the Fig. seems a better term?

Fig. 9: what is plotted along the horizontal axis? Time?

23, 539:       the occurrence of clouds was not mentioned in Sect. 3.2: which day was it? Was this the reason that Fig.7 shows only 4 cloudless days?

23, 564:       homogeneity: do you mean harmonization of procedures, recommendations for cloud screening, trace gas corrections, calibration procedures, etc?

---

## Referee Comment (RC2) · Anonymous Referee #2 · 3 Jan 2018

This manuscript by Kazadzis et al., "Results from the 4th WMO Filter Radiometer Comparison for aerosol optical depth measurements", presents the results of the 4th Filter Radiometer Comparison of AOD measurements held in Davos, Switzerland, where thirty filter radiometers and spectroradiometers from 12 countries participated.

The scope of the paper is both concise and specific. Moreover, the manuscript is clear and well written, so I have only some minor comments to be considered in the revised version before the publication.

General comment:

————————————-

[Figure]

Regarding the Figure 2, I was wondering that it would be perhaps interesting to focus on and discuss the diurnal patterns a bit more. For instance, POM_JP shows a diurnal pattern that is likely solely related to the calibration (as discussed in Cachorro et al. 2004). Or what do you think? Of course this would be more obvious to see and confirm, if the time in x-axis was a local solar time (instead of UTC time) and if similar plot would show the average hourly pattern (mean or median in each hour of local solar time). If this particular effect (of calibration) could be isolated with the help of diurnal plot, then it would give, at least in a rough sense, also a better quantitative idea about the role of the other sources causing the differences between different measurements.

By the way, in this figure the labels are not visible in the paper version, only if heavily zoomed in in the pdf-version (but this particular case seemed to be POM_JP). And the very small font size seemed to be a problem with the other figures as well.

Specific comments:
* * *
Line 48: I did not find these references as best suited here, for satellite-based AOD. Would some latest references for MODIS, MISR AOD algorithms and product perhaps be better fitted here?

Line 65: This sentence, regarding the references, was not immediately clear. What Neckel and Labs refers to, if given separately at the end of the sentence and other references earlier referring to the uncertainty estimate of 1%.

Line 429: "18% in common ...", should this be 0.8%?

Figure 9: Include the label for x-axis in this plot.

Line 531 (also in the line 38). I was thinking whether the word "sensitivity" is the best one here to give surely the right idea, idea being that the uncertainty in AE increases with decreasing AOD. Could one say that the sensitivity of AE to AOD then decreases as well? If so, is there a danger that one could misunderstand a statement like "sensitivity of this parameter at low AOD conditions".

Cachorro, V. E., P. M. Romero, C. Toledano, E. Cuevas, and A. M. de Frutos (2004), The fictitious diurnal cycle of aerosol optical depth: A new approach for "in situ" calibration and correction of AOD data series, Geophys. Res. Lett., 31, L12106, doi:10.1029/2004GL019651.

---

## Referee Comment (RC3) · Anonymous Referee #3 · 5 Jan 2018

General comments:

This manuscript well illustrates the great significance of aerosol optical depth and comparison results among 30 participated instruments from 12 countries in the 4th FRC. The high accuracy of reference aerosol network instruments demonstrated a promising framework to achieve homogeneity of AOD measurements on a global scale among the different spectral AOD networks in the near-future, which is interesting and attractive worldwide. It is scientifically well written and concise. I recommend to publish for ACP after minor revisions as followed.

Specific comments:

1. Line 50, it's better to change "ground based" to "ground-based". 2. Line 65, please don't use acronym when it first come across (i.e. UVA, NIR). 3. Line 154-155, Please check the English grammar. 4. Line 161, the unit "40 C" is incorrect, please check. 5. Line 175, Please check the English grammar "These types of comparisons are essential to moving. . ." 6. Line 528, delete the "s" at the end of "demonstrates". 7. Line 544, I suppose it should be "too much conservative cloud screening".

---

## Author Comment (AC2) · 23 Jan 2018

We thank the reviewer for his/her comments.

2, 49 (page, line): what is meant with 'the least common denominator' in this context? I think it is very common to use AOD in such comparisons.

Sentence corrected.

3, 67: may be good to define 'air mass'? What is 'relative air mass'?

Defined

4, 114: methods for

Corrected

5, 126: figure caption: not sure whether WORCC triad is common knowledge: suggest to refer to the text for explanation

Corrected to: Three reference PFR instruments (WORCC WORCC triad)

6, last para: In the routine handling of instruments in operational networks, also the maintenance and transport and installation are important factors which may influence the results. Are on‐site procedures to check the instruments after installation part of the routine? Are traveling standards used and site visits to check on procedures and maintenance? See also what is written on manual adjustment on 10, 285: is this done at the operational sites too?.

A text was added: "It has to be noted that most of the instruments have been installed, maintained and checked from the initial instrument operators that have been participated in the campaign. With the exception of two CIMEL instruments that PMOD WRC stuff has installed and maintained during the campaign. "

So each individual group has installed and maintained their instruments with the exception of two CIMEL instruments, which were sent and followed on line by the operators. The installation of the instruments has been performed by PMOD WRC stuff. This is more or less an easy task as there is a CIMEL already running at the site and PMOD stuff has experience on installing, maintaining the instrument using the experience of various international campaigns that has participated outside the home site using CIMEL instrumentation.

7, 184: instrumentation characteristics, calibration strategies and processing algorithms: is this info available for each network, and if so, would it be possible to provide a table where the most recent info can be obtained (may be the networks websites?).

We added relevant hyperlinks in the supplement related table.

7, 195: Chylek seems to be a rather old and may be a ran-
domly selected reference, I believe that the most recent version of
the GCOS requirements is more relevant for satellite measurements
(https://public.wmo.int/en/programmes/global‐climate‐observing‐system).

References have been corrected Levy, 2013, Sayer et al., 2012; Kahn et al., 2005; Li
et al., 2014 Toledano et al., 2011

9, 223: has instead of have?

Corrected

9, 233: remove comma after method,

Corrected

9, 251: Insert comma after USA

Corrected

10, 267: This comparison, what does 'this' refer to?;

Corrected

10, 268: separated in groups of different instrument types? (add groups of?)

Corrected

10, 274: the shorter wavelengths are not shown (unless I miss them, see my general
comment on figure quality).

Yes they are not shown in this figure. Added (not shown here). And figure quality has
been improved.

10 279: why does 'this prove the high level of the quality of reference instruments'?

Corrected to: "These results demonstrate the high level of the quality of reference
instruments belonging to the GAW-PFR. . ."

10, 283: on other days (add 'on')

Corrected

Figure 2: I cannot discriminate well between the colours but it seems to me that the 3 PFRs mentioned at the top of each legend are the same in each of the 6 plots, as well as the triad. Are these 3 PFRs together the triad? Is that why they are shown each time? And if so, why is the triad shown as a separate item?

Yes they were the triad and have been removed from the new figure version

15, 365: the lower the wavelength, the lower the reliability: is there an explanation for this? May be the increased Rayleigh scattering at lower wavelengths? Or molecular absorption?

It is difficult to generalize for all instruments. But in principle the increase in Rayleigh scattering (both in measurements and calibration related procedures), the lower irradiance signal are two possible reasons for this. However, each instrument has difference in its characteristics and characterization procedures so it is not correct to state something like this.

16, 381: As an example, for AODs . . .

Corrected

17, 389: suggest: differences using measurements form a three . . . .

corrected

17, 391: could lead to large deviations: what do you mean with that? Do you mean that you'd miss the higher AOD cases?

Changed to: "The use of such algorithms can lead to significant differences, while the selection of threshold values to filter out the retrievals could lead to large deviations comparing AOD retrievals from instruments with different cloud flagging algorithms."

Yes there are cases that very strict cloud flagging algorithms would miss higher AOD cases and also when very tolerant ones that lead to the inclusion of cloud "contaminated" (calculating high AOD) measurements.

18, 427 and 19, 452: not every minute? On 6, 157 (bullet c) is mentioned that CIMEL measures within a few seconds. Was CIMEL Used in it's routine operational mode, i.e. every 15 min? Or was it adjusted to continuous measurements to match the other instruments in the intercomparison? May be a few words should be said about his in the text?

The CIMEL instruments participated in the campaign have been measuring with a frequency of one measurement every three minutes. This information was added to the document.

Fig. 7, caption: four cloudless days? How does that compare with 5, 129?

Corrected

19, 443: what do you mean with 'artificial AOD time series' (also in legend of Fig. 8)? In the caption of Fig. 8 you call them mean AOD, which looking at the Fig. seems a better term?

In the text it is defined as : "This was constructed by spline-interpolating the mean AOD of all the remaining (three) instruments (excluding the CIMEL that has a lower temporal measurement frequency than the rest of the instruments), at the time intervals where the fourth instrument (SPO in this example) provides cloud free data." So for figure 7 is a mix of mean AOD and spline – interpolated AOD. For figure 8 is what it is defined the mean of three instruments.

Fig. 9: what is plotted along the horizontal axis? Time?

Yes. Corrected.

23, 539: the occurrence of clouds was not mentioned in Sect. 3.2: which day was it?

Was this the reason that Fig.7 shows only 4 cloudless days?

The reason for using 4 instead of 5 days in this figure was the fact that one out of five days used for the comparison was 100% cloudless with no hint of cirrus or any other cloud type, where all instruments mostly did not detect any clouds. Figure 8 shows only one day where cirrus clouds were present in certain times during the day which is tough test for all cloud flagging algorithms shown here.

23, 564: homogeneity: do you mean harmonization of procedures, recommendations for cloud screening, trace gas corrections, calibration procedures, etc?

Yes that is a much better statement, corrected to: "..starting point for global AOD harmonization of procedures, recommendations for cloud screening, trace gas corrections, calibration procedures."

---

## Author Comment (AC3) · 23 Jan 2018

We thank the reviewer for his/her comments.

Regarding the Figure 2, I was wondering that it would be perhaps interesting to focus on and discuss the diurnal patterns a bit more. For instance, POM_JP shows a diurnal pattern that is likely solely related to the calibration (as discussed in Cachorro et al. 2004). Or what do you think? Of course this would be more obvious to see and confirm, if the time in x-axis was a local solar time (instead of UTC time) and if similar plot would show the average hourly pattern (mean or median in each hour of local solar time). If this particular effect (of calibration) could be isolated with the help of diurnal

plot, then it would give, at least in a rough sense, also a better quantitative idea about the role of the other sources causing the differences between different measurements.

We added the following paragraph: Looking at possible diurnal patterns of the AOD differences shown in figure 2, most of the instruments show relatively constant differences over time (and air mass). One example of a possible diurnal pattern on the AOD differences that can be linked with the instrument calibration (as discussed in Cachorro et al., 2004) is the POM_JP instrument. There, differences are proportional to the 1/m and are up to 0.01 for high air masses. In this case if the calibration effect is isolated, the error on the instrument calibration (assuming the PFR triad calibration as ideal) is in the order of 1.6%.

By the way, in this figure the labels are not visible in the paper version, only if heavily zoomed in in the pdf-version (but this particular case seemed to be POM_JP). And the very small font size seemed to be a problem with the other figures as well.

Font sizes have been improved.

Line 48: I did not find these references as best suited here, for satellite-based AOD. Would some latest references for MODIS, MISR AOD algorithms and product perhaps be better fitted here?

References removed and added: Levy, 2013, Sayer et al., 2012; Kahn et al., 2005; Li et al., 2014 Toledano et al., 2011

Line 65: This sentence, regarding the references, was not immediately clear. What Neckel and Labs refers to, if given separately at the end of the sentence and other references earlier referring to the uncertainty estimate of 1%.

Corrected.

Line 429: "18% in common ...", should this be 0.8%?

It is actually 18% because it is the sum three percentages ( POM vs any of the other 3

instiments).

Figure 9: Include the label for x-axis in this plot.

Corrected

Line 531 (also in the line 38). I was thinking whether the word "sensitivity" is the best one here to give surely the right idea, idea being that the uncertainty in AE increases with decreasing AOD. Could one say that the sensitivity of AE to AOD then decreases as well? If so, is there a danger that one could misunderstand a statement like "sensitivity of this parameter at low AOD conditions".

Changed to: This was largely related to the uncertainty of this parameter that is linked with very low AOD uncertainties, at low AOD conditions.

---

## Author Comment (AC4) · 23 Jan 2018

We thank the reviewer for his/her comments.

1. Line 50, it's better to change "ground based" to "ground-based".

Corrected

2. Line 65, please don't use acronym when it first come across (i.e. UVA, NIR).

Corrected

3. Line 154-155, Please check the English grammar.

[Figure]

Corrected

4. Line 161, the unit "40 C" is incorrect, please check.

corrected

5. Line 175, Please check the English grammar "These types of comparisons are essential to moving: : :"

Corrected to: "These types of comparisons are essential in order to try to move forward the frontiers of instrument and metrology science."

6. Line 528, delete the "s" at the end of "demonstrates".

Corrected

7. Line 544, I suppose it should be "too much conservative cloud screening".

Corrected
* * *

---

## Author Comment (AC1)

**Results from the 4[th] WMO Filter Radiometer Comparison for aerosol optical depth measurements**

Stelios Kazadzis[1,11], Natalia Kouremeti[1], Henri Diémoz[2], Julian Gröbner[1], Bruce W. Forgan[3], Monica  Campanelli[4], Victor Estellés[5], Kathleen Lantz[6], Joseph Michalsky[6], Thomas Carlund[7], Emilio Cuevas[8], Carlos Toledano[9], Ralf Becker[10], Stephan Nyeki[1], Panagiotis G. Kosmopoulos[11], Viktar Tatsiankou[12], Laurent Vuilleumier[13], Frederick. M. Denn[14], Nozomu Ohkawara[15], Osamu Ijima[14], Philippe Goloub[16], Panagiotis I. Raptis[11,1], Michael Milner[3], Klaus Behrens[4], Africa Barreto[9,10,17], Giovanni Martucci[13], Emiel Hall[6], James Wendell[6], Bryan E. Fabbri[14], Christoph Wehrli[1]

[1]Physikalisch-Meteorologisches Observatorium Davos, World Radiation Center, Switzerland

[2]Aria e Atmosfera - Radiazione solare e atmosfera ARPA Valle, Saint-Christophe 11020, Italy

[3]Standards & Metrology, Bureau of Meteorology, Docklands Vic 3008, Australia

[4]SACI-CNR,Via Fosso del Cavaliere 100, 00133, Rome, Italy

[5]Department of Earth Physics and Thermodynamics, Solar Radiation and Research Unit, Univ. de València, Valencia, Spain

[6]Cooperative Institute for Research in Environmental Studies, NOAA/ESRL/GMD, Boulder, CO 80305, U.S.A.

[7]Swedish Meteorological and Hydrological Institute, 601 76 Norrköping, Sweden

[8]Izaña Atmospheric Research Centre, State Meteorological Agency (AEMET), Santa Cruz de Tenerife, Spain

[9]Atmospheric Optics Group (GOA), University of Valladolid 47011, Valladolid, Spain

[10]Deutscher Wetterdienst Meteorologisches Observatorium Lindenberg, D-15848 Tauche, Germany

[11]Institute of Environmental Research and Sustainable Development, National Observatory of Athens, Greece

[12] COFOVO Energy Inc. 800 King Edward Avenue, Suite 3014 Ottawa, ON, K1N 6N5 Canada

[13] Federal Office of Meteorology and Climatology MeteoSwiss, Payerne, Switzerland

[14] Science Systems & Applications Inc NASA Langley Science Directorate, Hampton VA 23666, U.S.A

[15] Japan Meteorological Agency 1-3-4 Otemachi, Chiyoda-ku, 100-8122 Tokyo, Japan

[16] Laboratoire d' Optique Atmosphérique, Univ. des Sciences et Technologies de Lille 159655 Villeneuve d'Ascq, France

[17]Cimel Electronique, 75011, Paris, France